# Cloning and Organelle Expression of Bamboo Mitochondrial Complex I Subunits Nad1, Nad2, Nad4, and Nad5 in the Yeast *Saccharomyces cerevisiae*

**DOI:** 10.3390/ijms23074054

**Published:** 2022-04-06

**Authors:** Hsieh-Chin Tsai, Cheng-Hung Hsieh, Ching-Wen Hsu, Yau-Heiu Hsu, Lee-Feng Chien

**Affiliations:** 1Department of Life Sciences, National Chung Hsing University, Taichung 402, Taiwan; aqulia1983@hotmail.com (H.-C.T.); topkofaxion7671@gmail.com (C.-H.H.); sandra258147@hotmail.com (C.-W.H.); 2Graduate Institute of Biotechnology, National Chung Hsing University, Taichung 402, Taiwan; yhhsu@nchu.edu.tw

**Keywords:** bamboo, mitochondrial complex I, proton pumping subunits, organelle expression

## Abstract

Mitochondrial respiratory complex I catalyzes electron transfer from NADH to ubiquinone and pumps protons from the matrix into the intermembrane space. In particular, the complex I subunits Nad1, Nad2, Nad4, and Nad5, which are encoded by the *nad1*, *nad2*, *nad4*, and *nad5* genes, reside at the mitochondrial inner membrane and possibly function as proton (H^+^) and ion translocators. To understand the individual functional roles of the Nad1, Nad2, Nad4, and Nad5 subunits in bamboo, each cDNA of these four genes was cloned into the pYES2 vector and expressed in the mitochondria of the yeast *Saccharomyces cerevisiae*. The mitochondrial targeting peptide *mt* gene (encoding MT) and the *egfp* marker gene (encoding enhanced green fluorescent protein, EGFP) were fused at the 5′-terminal and 3′-terminal ends, respectively. The constructed plasmids were then transformed into yeast. RNA transcripts and fusion protein expression were observed in the yeast transformants. Mitochondrial localizations of the MT-Nad1-EGFP, MT-Nad2-EGFP, MT-Nad4-EGFP, and MT-Nad5-EGFP fusion proteins were confirmed by fluorescence microscopy. The ectopically expressed bamboo subunits Nad1, Nad2, Nad4, and Nad5 may function in ion translocation, which was confirmed by growth phenotype assays with the addition of different concentrations of K^+^, Na^+^, or H^+^.

## 1. Introduction

Mitochondrial complex I (EC 1.6.5.3), also called NADH:ubiquinone (UQ) oxidoreductase, is the largest and most complicated enzyme complex in the respiratory electron transfer chain. Mitochondrial complex I catalyzes the transfer of two electrons from NADH to UQ, coupling the translocation of four protons across the inner membrane, thus contributing to the electrochemical potential [1,2]. All of the subunits of mitochondrial complex I have homologs in plants, animals, and bacteria [3,4,5,6,7]. Electron microscopy images have demonstrated that the structure of mitochondrial complex I in both animals and plants is L-shaped [3,8,9,10,11,12] and consists of a membrane arm and a peripheral arm: the membrane arm is embedded within the inner mitochondrial membrane and the peripheral arm protrudes into the matrix. The membrane arm subunit homologs in plants, animals, and bacteria are Nad1/ND1/NuoH/Nqo8, Nad2/ND2/NuoN/Nqo14/, Nad3/ND3/NuoA/Nqo7, Nad4/ND4/NuoM/Nqo13, Nad4L/ND4L/NuoK/Nqo11, Nad5/ND5/NuoL/Nqo12, and Nad6/ND6/NuoJ/Nqo10, which are encoded by mitochondrial DNA (mtDNA) or bacterial homologs [3,4,5]. In particular, Nad1, Nad2, Nad4, and Nad5 are reported to be involved in proton translocation [11,13,14].

In the human mitochondrial genome, there are six genes encoding complex I polypeptides, whereas in plants there is greater complexity in the mtDNA genes encoding complex I polypeptides [5,15,16,17]. The mtDNA gene content and genome size have greater variety among eukaryotic plants, though the core subsets of genes are present in all species. The genes of animal mtDNA do not contain introns and are sequential, whereas the genes of plant mtDNA contain introns and are fragmented. Compared with the mtDNA sequences found in other monocots [15,18,19], the homologous genes in the bamboo *Bambusa oldhamii* were encoded by distinct sets of exons [20]. Therefore, cDNAs that were reverse-transcribed from the fully edited mRNAs of *nad1*, *nad2*, *nad4*, and *nad5* from the two bamboo species *B. oldhamii* and *Phyllostachys edulis* were investigated in this study. Because an individual subunit may be encoded by several fragmented segments in the mitochondrial genome in higher plants, and cytidine (C) to uridine (U) RNA editing may occur randomly during mitochondrial transcription [21,22,23], the polypeptide sequences of the Nad1, Nad2, Nad4, and Nad5 subunits in mitochondrial RNA (mtRNA) sequences were also analyzed.

It is known that mutations in the ND1, ND2, ND4, and ND5 subunits can cause complex I dysfunction and result in mitochondrial diseases in humans [24,25,26]. In an *Arabidopsis otp43* mutant, the loss of the Nad1 subunit resulted in a lack of mitochondrial complex I activity and severe defects in seed development and germination [27]. In terms of functional roles, the plant Nad1 subunit has shown UQ binding and proton translocation activity [28,29]. With respect to the Nad2 subunit, Sato et al. [30] reported that NuoN (Nad2) in *Escherichia coli* may act as an antiporter-like subunit during energy transduction. Regarding Nad4, in a tobacco (*Nicotiana sylvestris*) mutant, the mutation of the Nad4 subunit leads to a loss of complex I assembly and lowering of leaf photosynthesis [31,32]. The sequence of the bacterial NuoM (Nad4) subunit has shown similarities to the Na^+^ or K^+^/H^+^ antiporter family, so NuoM (Nad4) has been proposed to participate in proton translocation [33]. As for Nad5, an important structural role for the C-terminal segments of Nad5 (NuoL) has been demonstrated. The shortening of the transmembrane helix of the C-terminal end of NuoL (Nad5) was shown to reduce the stability of the membrane arm of complex I, and the deletion of the C-terminal segments caused a total loss of the NADH-UQ oxidoreductase activity [34]. Further, a lack of the NuoL (Nad5) domain or parts of its C-terminal end reduced H^+^/2e^−^ stoichiometry in a ΔNuoL (Nad5) *Escherichia coli* mutant and indicated the critical role of this C-terminal helix in energy transduction [35]. The physiological roles of the plant mitochondrial Nad5 and Nad2 subunits are likely to involve proton transport in parallel with the *E. coli* NuoL and NuoN domains, which are multi-subunit Na^+^/H^+^ antiporters implicated in proton translocation [33].

It has been reported that the plant mitochondrial proteome contains at least 2000 nuclear gene products but only approximately 40 mitochondrial gene products [17]. Mitochondrial proteins containing MT peptides are encoded by nuclear genes, translated and synthesized in the cytosol, and then imported into mitochondria. The MT peptides are usually 20 to 40 amino acid residues long [36] and contain positively charged amino acid residues characteristically distributed along the sequence [37]. MT peptides contain the necessary information for directing mitochondrial preproteins to mitochondria. Therefore, in order to express bamboo Nad1, Nad2, Nad4, and Nad5 subunits in yeast mitochondria, it is necessary to fuse a suitable MT peptide at the N-terminal end of these polypeptides to direct them to the mitochondria. The ATPase δ subunit of *S. cerevisiae* is a nuclear-encoded polypeptide and contains an MT peptide of 22 amino acids [38]. The same ATPase δ subunit MT peptide has been used to direct the human ND5 (Nad5) subunit of complex I to yeast mitochondria [39] and was therefore chosen in this study.

Plant mitochondrial complex I is more complicated than its counterparts in other eukaryotes, such as humans. The study of the membrane subunits of mitochondrial complex I remains difficult due to their hydrophobicity and low natural abundance. Unlike other eukaryotic cells, the *S. cerevisiae* lacks mitochondrial complex I because it does not have genes homologous to the Nad1–Nad6 subunits present in the mitochondrial genome, and complex I cannot be assembled without those subunits [40]. *S*. *cerevisiae* has rotenone-insensitive NADH dehydrogenases that localize on the internal (Ndi1) and external (Nde1 and Nde2) faces of the inner mitochondrial membrane while these enzymes serve as alternatives to respiratory complex I in catalyzing NADH oxidation [41,42,43,44]. Nevertheless, *S*. *cerevisiae* is still an appropriate model organism to study the functional role of the individual complex I subunits [45,46]. A previous study has shown that the expression of the human ND5 subunit in *S. cerevisiae* was able to alter cation homeostasis [39] and illustrated the suitability of this approach.

Bamboo, which is an important Asian economic crop, serves as both a vegetable and a raw material in manufacturing. Our previous results [47] showed that complex I in both *Bambusa oldhamii* and *Phyllostachys edulis* behaves differently from other plants during metabolic energy transduction. Hence, we have attempted to subcellularly express the Nad1, Nad2, Nad4, and Nad5 subunits of bamboo mitochondrial complex I in *S. cerevisiae* to develop a model for future investigations.

## 2. Results

### 2.1. cDNA Alignments of nad1, nad2, nad4, and nad5 from B. oldhamii and P. edulis with Other Monocots

Total mtRNA was isolated from crude isolated bamboo mitochondria. The quality of mtRNA was checked with 1% (*w*/*v*) agarose gel electrophoresis (data not shown). The bamboo mtRNA samples were therefore subjected to reverse transcription (RT)-PCR to prepare the cDNAs, which were used as the templates for subsequent sequencing via the designed primers (Appendix A). Sequencing results from the *nad1*, *nad2*, *nad4*, and *nad5* cDNA fragments were obtained from the Biotechnology Center of NCHU and then aligned and assembled with BioEdit to generate the full-length genes. Full-length *nad1*, *nad2*, *nad4*, and *nad5* cDNA sequences of *B. oldhamii* (EU414244, KF219802, EU414245, and EU414246; Appendix A) and *nad1*, *nad4*, and *nad5* cDNA sequences of *P. edulis* (EU009478, EU414247, and EU414248; Appendix A) were submitted to the National Center for Biotechnology Information (NCBI) database.

The alignments of the *nad1*, *nad2*, *nad4*, and *nad5* cDNA were compared to the mitochondrial genomic DNA (mtDNA) sequences from the other monocots, including rice *Oryza sativa* (O.s.; DQ167399), wheat *Triticum aestivum* (T.a.; AP008982), and maize *Zea mays* (Z.m.; AY506529), as well as the sequence from *B. oldhamii* mtDNA (B.o.; EU365401) [20], in accordance with the NCBI database (Appendix A). The similarities between the *nad1*, *nad2*, *nad4*, and *nad5* cDNA genes of bamboo and the mtDNA of the other monocots (rice, maize, and wheat) were about 96.2%, 71.9%, 97.4%, and 98.6%, respectively [15,18,19]. Moreover, the similarities of the *nad1*, *nad4*, and *nad5* cDNA genes between *B. oldhamii* and *P. edulis* were about 99.8%, 99.9%, and 99.9%, respectively. It was not possible to compare the similarity between the *nad2* cDNAs of two bamboo species because the *nad2* cDNA of *P. edulis* could not be obtained for unknown reasons. In addition, the *nad1*, *nad2*, *nad4*, and *nad5* cDNAs of *B. oldhamii* had 98.5%, 73%, 99.3%, and 99.7% identity, respectively, with their mtDNA.

### 2.2. Polypeptide Alignment of Nad1, Nad2, Nad4, and Nad5 from B. oldhamii and P. edulis with Other Species

The Nad1, Nad2, Nad4, and Nad5 polypeptide sequences from *B. oldhamii* (B.o.; ABZ79354, KF219802, ABZ79355, and ABZ79356) and Nad1, Nad4, and Nad5 sequences from *P. edulis* (P.e.; ABV68729, ABZ79357, and ABZ79358) were translated from the cDNA sequences of the *nad1*, *nad2*, *nad4*, and *nad5* genes using BioEdit and then submitted to the NCBI database. The *B. oldhamii* Nad1, Nad2, Nad4, and Nad5 polypeptide sequences were aligned with *E*. *coli* (E.c.), *Homo sapiens* (H.s.), *Paracoccidioides brasiliensis* (P.b.), *Arabidopsis thaliana*, (A.t.), *O. sativa* (O.s.), *T*. *aestivum* (T.a.), and *Z*. *mays* (Z.m.), as well as the sequence translated from *B. oldhamii* mtDNA (B.o.; as shown in Figure 1).

It was found that many of the amino acid sequences of Nad1, Nad2, Nad4, and Nad5 in *B. oldhamii* and Nad1, Nad4, and Nad5 in *P. edulis* were the same as those in *A. thaliana*, *O*. *sativa*, *T. aestivum*, and *Z. mays*, but not in *E. coli*, *H. sapiens*, and *P. brasiliensis*. This compromised the observation that C-to-U RNA editing frequently occurred during polypeptide translation from the plant mitochondrial genomic DNA [23]. Surprisingly, the translated amino acid sequences of Nad1, Nad4, and Nad5 in *B. oldhamii* are nearly identical to those in *P. edulis*. There was only one different amino acid in Nad5 between *B. oldhamii* and *P. edulis*, with *B. oldhamii* Nad5 having Q615, which is the same as other monocotyledons, and *P. edulis* Nad5 having K615. The *B. oldhamii* Nad2 protein sequence was highly conserved (above 95.7% conservation) relative to the *B. oldhamii* mtDNA translated protein sequence [20]. The remaining 4.3% underwent C-to-U RNA editing in *B. oldhamii* Nad2; for example, 27S→27F, 103S→103F, etc.

### 2.3. Hydropathy Plots of Nad1, Nad2, Nad4, and Nad5

The hydropathy plots of Nad1, Nad2, Nad4, and Nad5 were created using BioEdit with a scan-window size of 19 (Figure 2) [48]. The homologues from different species had different polypeptide sequences but had a similar tendency towards hydrophobicity. The hydropathy plots showed that the numbers of transmembrane domains in Nad1, Nad4, and Nad5 from *B. oldhamii* and *P. edulis* were eight, fourteen, and seventeen, respectively. The number of transmembrane domains in Nad2 from *B. oldhamii* was twelve. The transmembrane domains in the Nad1, Nad2, Nad4, and Nad5 subunits from *B. oldhamii* and *P. edulis* were predicted with the TMHMM 2.0 program (https://services.healthtech.dtu.dk/service.php?TMHMM-2.0, accessed on 8 July 2021) and compared to the topologies of *E. coli* subunits (as shown in Appendix A) [37,49,50]. Because of the similar tendencies in the hydropathy plots of Nad1, Nad2, Nad4, Nad5, and their homologs, a similarity in transmembrane domain composition can be expected. As shown in Figure 1, the changes in amino acids, which probably resulted from C-to-U RNA editing, were F→L, H→Y, P→L, P→S, R→C, R→W, S→F, S→L, and T→I, and are marked with circles or triangles. These alterations may increase the hydrophobicity of the polypeptides, but those that resulted from L→F, as well as CCC→CCT and CTC→CTT, do not. The sequences of Nad1, Nad2, Nad4, and Nad5 subunits translated from cDNA were generally consistent with those translated from *B. oldhamii* mitochondrial genomic mtDNA (EU365401.1) [20]. However, the transmembrane domain distributions in the polypeptides translated from cDNA were different from those translated from genomic mtDNA. It is presumed that these C-to-U substitutions would cause amino acid changes and affect protein folding in the mitochondrial membrane.

The topology model of Nad1 (Figure 3A) suggested that the UQ binding site is probably located between segment IV, V, and VI, in which E148, E197, and E232 (Figure 3A) have been reported to be related to UQ reduction kinetics [49]. The transmembrane domains of Nad2 (Figure 3B) have been reported to have a similar antiporter structural feature to those of ND4 (Nad4) and ND5 (Nad5) and to be directly involved in H^+^ translocation [30]. No charged amino acids resulting from C-to-U substitutions existed that would affect the function of Nad2. In Figure 3C, the conserved amino acids in the Nad4 subunit and its homologous NuoM (Nad4) from *E. coli* were used to adjust the orientation of the Nad4 transmembrane domains [50]. Moreover, E147 and K227 (Figure 3C) of NuoM (Nad4) were reported to participate in H^+^ translocation [50]. In Figure 3D, the transmembrane domains of Nad5 were predicted according to the model of Mathiesen and Hägerhäll [33]. Two α-helices between segments IX and X were predicted to lie on the membrane surface. In addition, there was one more α-helix at the Nad5 C-terminus, unlike other higher plant homologs. The region between segments IV to XII was conserved in all antiporter-like polypeptides.

### 2.4. Construction of pYES2-mt-egfp Plasmid-Carrying nad1, nad2, nad4, or nad5 Genes

The bamboo Nad1, Nad2, Nad4, and Nad5 subunits of mitochondrial complex I are encoded by mtDNA, synthesized by mitochondrial ribosomes, and do not possess MT peptides. In order to direct the bamboo Nad1, Nad2, Nad4, and Nad5 subunits to yeast mitochondria, an MT peptide (encoded by nuclear *mt* gene) of the ATPase δ subunit from *S. cerevisiae* [38] was employed to fuse at the 5′-terminal of the *nad1*, *nad2*, *nad4*, and *nad5* genes. This MT peptide should allow Nad1, Nad2, Nad4, and Nad5 to pass through the mitochondrial outer membrane and to be folded into the inner membrane [51]. Plasmids containing these genes were then constructed using the methods described in the Materials and Methods. To examine whether the *nad1*, *nad2*, *nad4*, and *nad5* genes were inserted correctly into pYES2-mt-egfp (Figure 4 and Appendix A), the plasmids were transformed into *E. coli* competent cells and the transformants were confirmed using colony PCR with a pair of primers, mt-egfp-F and mt-egfp-R (Appendix A). The expected 0.3 kb *mt-egfp*, 1.3 kb *mt-nad1-egfp*, 1.6 kb *mt-nad2-egfp*, 1.8 kb *mt-nad4-egfp*, and 2.2 kb *mt-nad5-egfp* PCR fragments were observed on an agarose gel (data not shown). The constructed plasmids were further confirmed by DNA sequencing and were found to be accurate after comparison with the cDNA sequences of the *nad1*, *nad2*, *nad4*, or *nad5* genes (data not shown).

### 2.5. Yeast Transformants

#### 2.5.1. Yeast Transformants Containing pYES2-mt-nad1-egfp, pYES2-mt-nad2-egfp, pYES2-mt-nad4-egfp, or pYES2-mt-nad5-egfp

The plasmids pYES2, pYES2-egfp, pYES2-nad1-egfp, pYES2-mt-egfp, pYES2-mt-nad1-egfp, pYES2-mt-nad2-egfp, pYES2-mt-nad4-egfp, or pYES2-mt-nad5-egfp (Figure 1) were transformed into *S. cerevisiae*. The yeast transformants were visualized on SC-U medium selective plates (Appendix A Appendix A and Appendix A). The yeast transformants containing the constructed plasmids were confirmed using colony PCR with a pair of primers, mt-egfp-F and mt-egfp-R (Appendix A). The expected 0.3 kb *mt-egfp* fragment was observed in MG transformants containing the pYES2-mt-egfp plasmid, the expected 1.3 kb *mt-nad1-egfp* fragment was observed in MN1G transformants containing the pYES2-mt-nad1-egfp plasmid, the expected 1.6 kb *mt-nad2-egfp* PCR fragment was observed in MN2G transformants containing the pYES2-mt-nad2-egfp plasmid, the expected 1.8 kb *mt-nad4-egfp* fragment was observed in MN4G transformants containing the pYES2-mt-nad4-egfp plasmid, and the expected 2.2 kb *mt-nad5-egfp* PCR fragment was observed in MN5G transformants containing the pYES2-mt-nad5-egfp plasmid (Appendix A).

#### 2.5.2. RNA Transcription of Fusion Genes in Yeast Transformants

Yeast total RNA isolated from the transformants containing the constructed plasmids was analyzed with agarose gel electrophoresis. The major 28S and 18S rRNA bands were recognized at 3 kb and 1.6 kb on the agarose gel (data not shown). The total RNA of yeast transformants was then reverse-transcribed to cDNA. The yeast transformant cDNAs were used as templates for PCR with several pairs of primers: mt-egfp-F and mt-egfp-R/*Bam*HI-nad2-F and *Bam*HI-nad2-R/*Eco*RI-nad5-F and *Xma*I-nad5-R (Appendix A). The expected 0.3 kb *mt-egfp* fragment was observed in MG (Figure 5A), the 1.3 kb *mt-nad1-egfp* fragment was observed in MN1G (Figure 5A), the 1.8 kb *mt-nad4-egfp* fragment was observed in MN4G (Figure 5A), the 1.5 kb *nad2* fragment was observed in MN2G (Figure 5B), and the 2.0 kb *nad*5 fragment was observed in MN5G (Figure 5B). This suggests that these fusion genes could be transcribed in yeast transformants.

#### 2.5.3. Expression of MT-Nad1-EGFP, MT-Nad2-EGFP, MT-Nad4-EGFP, or MT-Nad5-EGFP Fusion Proteins in Yeast Transformants

The total membrane proteins were isolated from the yeast wild-type (WT) and transformants MN1G, MN2G, MN4G, and MN5G. The membrane proteins were analyzed with 12% SDS-PAGE electrophoresis (Appendix A). The MT-EGFP, MT-NAD1-EGFP, MT-Nad2-EGFP, MT-Nad4-EGFP, and MT-Nad5-EGFP proteins were labeled with anti-GFP on PVDF membrane using Western blot analysis, and the 29 kDa MT-EGFP, 53 kDa MT-Nad1-EGFP, and 65 kDa MT-NAD4-EGFP were observed on the blots containing yeast total membrane proteins from MG, MN1G, and MN4G, respectively (Appendix A). Unfortunately, the signals from the 84 kDa MT-NAD2-EGFP and the 102 kDa MT-NAD5-EGFP were too weak to be visualized on blots containing either yeast total membrane proteins or mitochondrial proteins (data not shown).

#### 2.5.4. Subcellular Localization of MT-NAD1-EGFP, MT-NAD2-EGFP, MT-NAD4-EGFP, or MT-NAD5-EGFP Fusion Proteins in Yeast Observed Using EGFP Fluorescence

In order to investigate whether the fusion proteins were targeted to subcellular mitochondria, the EGFP fluorescence was detected in the yeast transformants MN1G, MN2G, MN4G, or MN5G using a fluorescence microscope. The yeast WT was used as a reference because it did not contain EGFP and did not emit green fluorescence. The yeast transformant containing MT-EGFP (MG) was used as a comparison for EGFP green fluorescence. The cells of the yeast WT and transformants were induced with SC-U medium plus 2% galactose for 0–9 h at 30 °C and then a time course of EGFP fluorescence development was observed with a fluorescence microscope. The images were taken at (A) 0 h, (B) 3 h, (C) 6 h, and (D) 9 h (Figure 6). Bright field microscopy was used to monitor the integrity of the cells. The yeast transformants containing MT-EGFP(MG), MT-Nad1-EGFP (NM1G), MT-Nad2-EGFP (NM2G), MT-Nad4-EGFP (NM4G), and MT-Nad5-EGFP (NM5G) had observable green fluorescence from the EGFP fusion protein and red fluorescence from MitoTracker Red staining (Figure 7), with fluorescence localized to the same green and red spots (yellow in the merged images) indicating the targeting of EGFP fusion proteins to the mitochondria. The yeast transformants containing EGFP only and Nad1-EGFP were used as the comparisons (Appendix A).

### 2.6. K^+^, Na^+^, or H^+^-Dependent Growth Phenotypes of Yeast Transformants

Since there were some reports showing that Nad1, Nad2, Nad4, or Nad5 subunits may not involve in direct proton translocation [49,52,53,54,55,56], the analysis of cation homeostasis is carried out to confirm whether the individual subunits of these four subunits in bamboo could function independently as ion translocators.

To analyze the ion translocations of yeast transformants, the yeast transformants were grown under different concentrations of K^+^, Na^+^, or H^+^. The yeast transformants MG, NM1G, NM2G, NM4G, and NM5G were grown on SC-U medium (Appendix A) plus 2% galactose plates with the addition of K^+^-containing solutions (0.8 M KCl and 1 M KCl), Na^+^-containing solutions (0.8 M NaCl and 1 M NaCl), or H^+^-containing solutions (pH 6.5, pH 5.5, and pH 4.5; Figure 8). The WT (Y) and transformants containing EGFP, Nad1-EGFP, and MT-EGFP were used as the comparisons (Appendix A). All of the yeast transformants were able to grow under conditions of 1 M KCl, 1 M NaCl, and pH 4.5. These results imply that the MN1G, MN2G, MN4G, and MN5G transformants can function independently from ion translocations.

## 3. Discussion

### 3.1. Highly Conserved Nad1, Nad2, Nad4, and Nad5 in Plants

The cDNAs of *B. oldhamii* and *P. edulis* were reverse-transcribed from mitochondrial total RNA and the cDNAs of *nad1*, *nad4*, and *nad5* from both *B. oldhamii* and *P. edulis*, as well as *nad2* from *B. oldhamii*, were sequenced and compared. The *nad1*, *nad4*, and *nad5* cDNA sequences showed more than 99.8% similarity with *B. oldhamii* and *P. edulis*. Compared to other monocotyledonous plants (including wheat, rice, and maize), the identities of *nad1*, *nad4*, and *nad5* cDNAs from *B. oldhamii* and *P. edulis* were still higher than 96.2%. In addition, the *nad2* cDNA sequence from *B. oldhamii* showed 71.9% nucleic acid identity to homologous sequences from other monocotyledons and Arabidopsis.

After comparison with wheat, rice, and maize mitochondrial genome maps, it was found that mitochondrial genes were highly conserved but there was little shared synteny between these species [18,19,57]. For instance, in the wheat mitochondrial genome map, *nad2* and *nad5* are divided into five exons, while in the *B. oldhamii* mitochondrial gene map, *nad2* is divided into five exons and *nad5* is divided into four exons [15,20,58]. Therefore, isolating mitochondrial mRNA was considered to be a better method for this study than isolating the entire mtDNA genome.

The mitochondrial total RNA, including mRNA, was reverse-transcribed into cDNA using random hexamers as primers, which gave better sequence preservation than oligo dTs because plant mitochondrial mRNA would be degraded by a 3′- to 5′-exoribonuclease if the mitochondrial mRNA had been polyadenylated. Additionally, non-AUG start codons have been found to initiate the translation of proteins in mitochondria and chloroplasts [59,60].

The alignment of cDNA and mtDNA sequences showed that there may be some C-to-U RNA editing sites in *nad1*, *nad2*, *nad4*, and *nad5*. These editing sites were found to be non-random and clustered in groups [61]. It was proposed that C-to-U RNA editing would result in correct expression and an increase in the hydrophobicity of the protein. It has been shown that RNA editing in mitochondria can result in the conservation of mitochondrial proteins among plants [62].

The cDNA and amino acid sequences of the bamboo Nad1 and Nad4 subunits were compared with the human ND1 (Nad1) and ND4 (Nad4) subunits [63]. The identities of cDNA and amino acid sequences between bamboo and human were 26% and 41%, respectively. However, the hydropathy plots of the bamboo Nad1 and Nad4 suggested that the corresponding subunits in humans have similar transmembrane topologies.

It was found in *nad1*, *nad4*, and *nad5* cDNA that there were a number of nucleotide differences, such as TTC→TTT, AAA→AAG, ACA→ACT, and TCC→TCA, that seemingly led to no change in amino acid residues during translation, apart from one difference—CAA→AAA (Appendix A). The *nad2* cDNA sequence of *B. oldhamii* (Appendix A) was compared to its mtDNA sequence showing that the *B. oldhamii nad2* cDNA sequence had 73% similarity with its mtDNA sequence, while the remaining 27% comprised C-to-T RNA editing. Many of these differences in cDNA led to no change during translation.

The polypeptide sequences and the hydropathy plots of Nad1, Nad4, and Nad5 suggested that these subunits may have similar structures in higher plants, mammals, bacteria, and fungi (Figure 2A,C,D). The highly conserved amino acids reported previously in the Nad1, Nad4, and Nad5 sequences of other species were also found in *B. oldhamii* and *P. edulis*. The amino acid sequences of the Nad1 and Nad4 subunits in *B. oldhamii* are also identical to those in *P. eduli**s*, while the Nad5 subunit in *B. oldhamii* is nearly identical to *P. edulis*, except for a single substitution of Q615 in *B. oldhamii* for the K615 in *P. edulis*. The polypeptide sequences and the hydropathy plots of Nad2 suggested that this subunit may have similar structures in higher plants, mammals, bacteria, and fungi (Figure 2B).

### 3.2. Transmembrane Domain Topology of Nad1, Nad2, Nad4, and Nad5

The Nad1, Nad4, and Nad5 amino acid sequences of *B. oldhamii* and *P. edulis*, as well as the Nad2 amino sequence of *B. oldhamii*, were translated from the cDNA sequences and analyzed with hydropathy plots and for transmembrane topology. Nad1 homologs have been reported to have eight transmembrane domains. Three conserved glutamates (E158, E212, and E247) have been located on the internal surface of the membrane in the bacterium *P. denitrificans*, and these were reported to be involved in binding and reducing UQ [49]. It has also been suggested that the mutagenesis of the ND1 (Nad1) subunit of complex I affects ubiquinone reduction kinetics [64]. The sequence alignments showed that these three conserved glutamates were also found in the bamboo Nad1 subunit (E148, E197, and E232). The simulations of transmembrane topology suggested that these conserved glutamates of Nad1 are located at the matrix side of the membrane. This implies that the domain between segments IV and VI of Nad1 in *B. oldhamii* and *P. edulis* might be involved in the binding and reduction in UQ. Lim et al. [65] have stated that a loss of Nad1 results in the disruption of complex I biogenesis during the early stages of assembly. Nad2 transmembrane topology was predicted and compared with the topology of its bacterial homolog NuoN (Nad2), which was originally reported by Tursun et al. [66]. It was suggested that there were important conformational changes in a specific region for the delivery of protons to the periplasm or for coupling the actions of subunit N (Nad2) with subunit M (Nad4). Nad4 transmembrane topology was predicted and compared with its bacterial homolog NuoM (Nad4). When comparing the sequences of *B*. *oldhamii* and *P*. *edulis* with other monocotyledons, there was one difference, with L347 in the transmembrane segment X of Nad4 in bamboo rather than the P347 present in other monocotyledons (in this study) [67]. In the case of Nad5, transmembrane topology was also predicted and compared with the topology of its homolog NuoL (Nad5) [33]. A comparison of the Nad4 and Nad5 sequences suggested that these two subunits participate in UQ binding and proton translocation in complex I [68,69,70]. The UQ binding motif could be located at H97, H239, H257, and H357 in the Nad5 subunits of *B. oldhamii* and *P. edulis*.

### 3.3. Allotopic Expression and Localization of Bamboo Nad1, Nad2, Nad4, and Nad5 in Yeast

The main subject addressed in this study was the expression of bamboo Nad1, Nad2, Nad4, and Nad5 subunits in the mitochondria of *S. cerevisiae*. A proposed approach is allotopic expression, in which the mtDNA-encoded gene is transferred to the nucleus and the protein is synthesized in the cytosol and subsequently imported into the mitochondria [71]. There are some proof-of-principle reports that have used such an allotopic approach to import wild-type ND1 (Nad1) or ND4 (Nad1) into mitochondria for treatment of a human metabolic disease [72,73]. In this study, in the absence of the MT peptide, the import of Nad1-EGFP into yeast mitochondria failed because the green fluorescence mistily appeared over the whole cell (Appendix A). The sorting mode was altered by adding an MT peptide, as in the case of the MT-Nad1-EGFP, MT-Nad2-EGFP, MT-Nad4-EGFP, or MT-Nad5-EGFP fusion proteins, which were directed toward *S. cerevisiae* mitochondria and colocalized with the MitoTracker Red mitochondrial marker probe (Figure 7). Additionally, the MT-EGFP, MT-Nad1-EGFP, and MT-Nad4-EGFP were detected with anti-EGFP in the portion of the total membrane proteins (Appendix A) and mitochondrial membrane proteins (Appendix A) using Western blot analysis. Therefore, it indicates that these proteins should be located on the mitochondrial membrane. Furthermore, these proteins carry MT peptide of ATP synthase δ subunit, which is a transmembrane protein in the mitochondrial inner membrane, so they should be led to the mitochondrial inner membrane via the TOM-TIM pathway [74,75,76].

The overexpression of Nad1, Nad2, Nad4, or Nad5 proteins in the inner mitochondrial membrane might affect the expression of other proteins, in particular, Ndi1, Nde1, and Nde2. It is known that these three enzymes are peripheral proteins on the surface of mitochondrial inner membranes that serve as an alternative respiratory complex I to catalyze the transfer of two electrons from NADH to UQ without proton translocation [41,42,43,44]. It was demonstrated that knockout of Nde1 or Nde2 in *S*. *cerevisiae* would cause a significant decrease in the NADH respiration rate of the cells [42]. Regarding this concern, it may require further experiments to confirm.

Actually, the effect of the heterologous expression of Nad1 and Nad4 subunits in yeast on the mitochondrial oxygen consumption rate, with succinate used as respiratory substrate, was studied. The data showed that the oxygen consumption rates of the yeast transformants containing Nad1 or Nad4 were significantly decreased as compared to the control (as shown in Appendix A). A possible mechanism could be that the folding of the foreign subunit tailed with EGFP in the mitochondrial inner membrane influences the spatial structure and electron transfer of the respiratory chain and results in the decrease in respiration rate.

### 3.4. Functional Characterization of the Bamboo Nad1, Nad2, Nad4, and Nad5 Subunits in Yeast Mitochondria

The Nad1, Nad2, Nad4, and Nad5 subunits are the core membrane subunits of the mitochondrial complex I [3]. These four subunits are intrinsic membrane proteins with several transmembrane helices (9 for Nad1, 14 for Nad2, 13 for Nad4, and 16 for Nad5) predicted by the TMHMM program (https://services.healthtech.dtu.dk/service.php?TMHMM-2.0, accessed on 8 July 2021). The importance of Nad1, Nad2, Nad4, and Nad5 in the assembly of mitochondrial complex I has been demonstrated through mutant studies. In mutants of complex I in the green alga *Chlamydomonas*, the absence of intact Nad1 affected the complete assembly of complex I [11,77]. Further, a number of Nad1 mutants in plants have been characterized [78] and it has been demonstrated that the deletion of the Nad1 subunit in the *Arabidopsis otp43* mutant results in a loss of complex I activity and a decrease in the respiration rate [27]. Nad2 acts as an antiport-like subunit, which may have the function of ion translocation or proton pumping. The Nad4 subunit leads to the formation of a reduced molecular weight subcomplex of 650 kDa in *Chlamydomonas* complex I mutants [77]. In another example, mutation of the Nad4 subunit in a maize mutant was associated with reduced complex I function [79]. Regarding Nad5, the deletion of the helix in the bacterial homolog NuoL reduced H^+^/e^−^ stoichiometry, indicating its direct involvement in proton translocation in complex I [80]. Other studies have proposed that the energy released by the redox reaction is transmitted to the conformational change in the NuoL horizontal helix [81] and that the C-terminal helix of the subunit plays an important role in energy transmission [39].

Overall, this study was successful in expressing the bamboo Nad1, Nad2, Nad4, and Nad5 subunits in yeast mitochondria and it has provided a possible pathway for further investigation of these membrane proteins.

## 4. Materials and Methods

### 4.1. Plant Materials and Microbes

Fresh edible bamboo rhizome shoots of *Bambusa*
*oldhamii* and *Phyllostachys edulis* were obtained from the local market. The leaf sheaths covering the shoots were removed, and only the top 10 cm of the shoot was used. *Escherichia coli* strain TOP10F’ (Invitrogen, now part of Thermo Fisher Scientific, Waltham, MA, USA) was used in all recombinant DNA experiments and grown in Luria-Bertanie (LB) broth or on LB agar plates at 37 °C. *Saccharomyces cerevisiae* INVSc1 (MATα, *his3Δ1*, *leu2*, *trp1-289*, *ura3-52*/MATα, *his3Δ1*, *leu2*, *trp1-289*, *ura3-52*; Invitrogen) was cultured in yeast extract–peptone–dextrose (YPD) medium at 30 °C.

### 4.2. Preparation of Mitochondria from Young Bamboo Shoots

Young bamboo shoots were freshly obtained from a local farm. Mitochondria were isolated according to the method of Chien et al. [47] and all processes were carried out at 4 °C. The crude mitochondria pellet was gently homogenized in the sterile solution to a final concentration of 400 mM mannitol, 20 mM 3-(N-morpholino)propanesulfonic acid (MOPS) (pH 7.2), 1 mM ethylene glycol-bis(β-aminoethyl ether)-*N*,*N*,*N*′,*N*′-tetraacetic acid (EGTA), 0.1% (*w*/*v*) BSA, and approximately 10 mg mitochondrial protein mL^−1^. The concentration of the crude mitochondrial protein suspension was determined by the Biuret Assay using BSA as a standard [82]. Isolated mitochondria were stored at −80 °C until use.

### 4.3. Isolation of Total Bamboo Mitochondrial RNA (mtRNA)

Total mtRNA was isolated by the phenol-chloroform method. About 10 mg of crude mitochondria were centrifuged at 4 °C for 15 min at 15,000× *g* (Hitachi CF15RX, Tokyo, Japan) to remove the supernatant. The pellet was resuspended gently with a mixture containing 250 μL of phenol and 250 μL of RNA extraction solution comprising 200 mM LiCl, 200 mM Tris-HCl (pH 8.0), 20 mM EDTA, and 0.2% (*w*/*v*) SDS. The RNA extract was incubated at 80 °C for 5 min. Then, 100 μL of chloroform and isopropanol (24:1) was added to the RNA extract and inversed gently. The mixture was centrifuged at 4 °C for 5 min at 12,000× *g*. The clear supernatant was transferred to a new Eppendorf tube, topped up with an equal volume of 4 M LiCl, and then incubated at −80 °C for at least 2 h. Then, the mixture was centrifuged at 4 °C for 20 min at 12,000× *g* and the white pellet settled at the bottom of the Eppendorf. After the supernatant was removed, the pellet was resuspended in 100 μL of sterilized ddH_2_O. Then, 10 μL of 3 M NaOAC (pH 5.2) and 250 μL of 100% ethanol were added to the suspension. The mixture was incubated at −70 °C for 2 h and then centrifuged at 4 °C for 15 min at 12,000× *g*. Subsequently, the RNA pellet was washed with 500 μL of 70% ethanol, centrifuged at 4 °C for 5 min at 12,000× *g*, and dried inside a laminar flow cabinet for 20 min. The RNA pellet was resuspended in 30 μL of sterilized ddH_2_O and stored at −70 °C.

### 4.4. cDNA Synthesis by the Reverse-Transcription Polymerase Chain Reaction (RT-PCR)

The RNA and dT mixtures comprised 2 or 5 μg of RNA template combined with 0.5 μL of 1 μg μL^−1^ (5 μM) random hexamer and ddH_2_O to a final volume of 12 μL. The RNA and dT mixtures were incubated at 70 °C for 50 min and then chilled on ice for 1 min. The mixtures were gently mixed with 7 μL of reaction buffer containing 1× FS of PCR buffer (Invitrogen), 2.5 mM MgCl_2_, and 0.5 mM dNTP. Then, the RNA and dT and reaction buffer mixtures were incubated at 42 °C for 5 min. They were further incubated at 42 °C for 50 min after adding 200 U superscript II reverse transcriptase (Sigma-Aldrich, now part of Merck, Darmstadt, Germany or Invitrogen, Waltham, MA, USA). The reaction was terminated at 70 °C for 15 min and the obtained cDNA was stored at −20 °C until later use.

### 4.5. Sequencing and Alignment

To sequence the *nad1*, *nad2*, *nad4*, and *nad5* cDNA genes from *B. oldhamii* and *P. edulis*, several pairs of forward and reverse primers (as shown in Appendix A) were designed with Primer3 [83] (http://primer3.wi.mit.edu/, accessed on 30 January 2015) in accordance with monocot mitochondrial DNA (mtDNA) sequences from *O*. *sativa* (O.s., DQ167399), *T*. *aestivum* (T.a., AP008982), and *Z*. *mays* (Z.m., AY506529). The primers were then used to amplify cDNA fragments by RT-PCR. The RT-PCR products were purified with a Gel Elution Kit (GeneMark, Taichung, Taiwan) and sent to the Biotechnology Center, National Chung Hsing University (Taiwan) for sequencing. The results were aligned and assembled using BioEdit software (http://www.mbio.ncsu.edu/BioEdit/, accessed on 24 September 2016; as shown in Appendix A). The Nad1, Nad2, Nad4, and Nad5 polypeptide sequences translated from *nad1*, *nad2*, *nad4*, and *nad5* cDNA sequences were aligned with those from *E*. *coli*, *H*. *sapiens*, *P*. *brasiliensis*, *A. thaliana*, *O*. *sativa*, *T. aestivum*, and *Z. mays* (as shown in Figure 1).

### 4.6. Plasmid Construction

The 1.0 kb *nad1* fragment and the 1.5 kb *nad4* fragment were amplified from *P. edulis* mitochondrial cDNA, and the 1.5 kb *nad2* fragment and the 2.0 kb *nad5* fragment were amplified from *B. oldhamii* mitochondrial cDNA using PCR with different pairs of primers (Appendix A). The restriction sites at the 3′ terminus were without stop codons (Appendix A). The PCR products were analyzed on agarose gels and purified from the gels with an elution kit (Favorgene, Pingtung, Taiwan; data not shown). The purified *nad1*, *nad2*, *nad4*, and *nad5* fragments were ligated into the yT&A (TA) vector as TA-nad1, TA-nad2, TA-nad4, and TA-nad5. The pYES2 (Invitrogen, Waltham, MA, USA) vector was used to express the bamboo genes. The sequence coding the *mt* from the *S. cerevisiae* ATPase δ subunit (5′ aagcttATGTTACGTTCAATTATTGGAAAGAGTGCATCAAGATCATTGAATTTCGTCGCTAAGCGTTCATATCACCACCATCACCATCACCATCACaagctt 3′) was synthesized (MDBio, Taipei, Taiwan). The *mt* and *egfp* genes were subsequently ligated into pYES2 (Figure 4) following the protocol provided with the Clone EZ kit (GenScript, Piscataway, NJ, USA). The *nad1* insert amplified with a designed primer set (Appendix A) was ligated into pYES2-egfp between *Hind*III and *Eco*RI sites by adding 5× DNA ligase reaction buffer and 1 U of T4 DNA ligase (Invitrogen) to yield pYES2-nad1-egfp. pYES2-mt-nad1-egfp was then obtained after the *mt* gene was subsequently ligated into the plasmid (Figure 4 and Appendix A). The *nad4* insert containing the *Kpn*I restriction site was cloned into pYES2-mt-egfp using the Clone EZ kit protocol (GenScript, Piscataway, NJ, USA). A 5 μL quantity of the *nad4* PCR product and 600 ng μL^−1^ of linearized pYES2-mt-egfp were added to 14 μL of reaction mixture containing 2 μL of 10× CloneEZ buffer and 1 U μL^−1^ of CloneEZ enzyme. After gently mixing by pipetting, the reaction mixtures were incubated at 22 °C for 30 min and incubated on ice for 5 min. pYES2-mt-nad4-egfp was then obtained (Figure 4 and Appendix A). The *B. oldhamii nad2* insert was prepared from TA-nad2 clones using *Bam*HI restriction digestion and ligated into multiple cloning sites of the pYES2-mt-egfp plasmid between *mt* and *egfp* by adding T4 DNA ligase to yield pYES2-mt-nad2-egfp (Figure 4 and Appendix A). Similarly, the *B. oldhamii nad5* insert was prepared from the TA-nad5 clone using *Eco*RI and *Xma*I restriction digestion and ligated into multiple cloning sites of the pYES2-mt-egfp plasmid between *mt* and *egfp* by adding T4 DNA ligase to yield the pYES2-mt-nad5-egfp (Figure 4 and Appendix A). All the constructed plasmids were confirmed by DNA sequencing (data not shown).

### 4.7. Yeast Transformation

Yeast transformation and yeast-competent cell preparation were carried out according to the methods of Gietz et al. [84] with some modifications, and all processes occurred at room temperature. A colony of *S. cerevisiae* INVSc1 (Invitrogen) was added to 3 mL of YPD medium (Appendix A) and cultured overnight at 30 °C. The cell concentrations were determined and diluted to an OD_600_ of 0.4 in 30 mL of YPD medium and incubated for an additional 2–4 h. The cells were centrifuged for 5 min at 3000× *g* and the cell pellets were resuspended in 15–20 mL 1× TE buffer containing 10 mM Tris-HCl (pH 7.5) and 1 mM EDTA. Then, the cell suspensions were centrifuged for 5 min at 3000× *g* and the pellets were resuspended with 1–2 mL of 1× LiAc/0.5× TE buffer containing 100 mM LiAc, 5 mM Tris-HCl (pH 7.5), and 0.5 mM EDTA. A 90 μL aliquot of yeast-competent cell suspensions was added to an Eppendorf tube containing 10 μL of glycerol and was quickly stored at −70 °C.

For yeast transformation, the yeast-competent cells were incubated at room temperature for 10 min. A 1 μg aliquot of pYES2-mt-egfp, pYES2-mt-nad1-egfp, pYES2-mt-nad2-egfp, pYES2-mt-nad4-egfp, or pYES2-mt-nad5-egfp plasmids was added to 100 μL of yeast-competent cells and 600 μL of LiAc/PEG/TE buffer containing 100 mM LiAc, 40% PEG-6000, 10 mM Tris-HCl (pH 7.5), and 1 mM EDTA, and incubated at 30 °C for 30 min. Then, 70 μL of DMSO was added, mixed, and incubated at 42 °C for 7 min. The mixture was centrifuged for 10 s at 15,000× *g* and the pellet was resuspended in 500 μL of 1× TE buffer. The cell suspension was centrifuged for 10 sec at 15,000× *g*, and the resulting pellets were resuspended in 100 μL of 1× Tris-EDTA(TE) buffer and then plated onto selective plates.

### 4.8. Induction of MT-EGFP, MT-Nad1-EGFP, MT-Nad2-EGFP, MT-Nad4-EGFP, and MT-Nad5-EGFP in Yeast

The yeast transformants were cultured in 3 mL SC-U medium (Appendix A) plus 2% glucose with shaking at 225 rpm overnight. Gene expression under the control of the inducible galactosidase (GAL) promoter was induced by adding galactose into the cell culture. The overnight cultures were harvested by centrifugation at 3000× *g* and this was followed by washing three times with 1 mL ddH_2_O. The cells were then resuspended in induction medium (SC-U plus 2% galactose; Appendix A). The OD_600_ of the cell suspensions was adjusted to 0.4 and the cell cultures were incubated at 30 °C for 8–9 h.

### 4.9. Isolation of Total RNA from Yeast

Yeast total RNA was prepared according to the method of Schmitt et al. [85] with modifications. Cell cultures were harvested by centrifugation at 3000× *g* for 5 min and the cell pellet was resuspended in 400 μL AE buffer containing 50 mM NaOAc (pH 5.5) and 10 mM EDTA. The cell suspension was transferred into a 1.5 mL microcentrifuge tube, 40 µL of 10% SDS was added, and then the mixture vortexed for 2 min. An equal volume of phenol was subsequently added to the tube, followed by vortexing and incubation at 65 °C for 4 min. The reaction mixture was rapidly chilled to −80 °C until phenol crystals appeared and was then centrifuged at 12,000× *g* for 2 min at 4 °C. A 400 μL aliquot of the aqueous phase solution was transferred to a new tube and an equal volume of phenol, chloroform, and isopropanol (25:24:1) was added, vortexed, and centrifuged at 12,000× *g* for 5 min at 4 °C. Again, a 200 μL volume of the aqueous phase solution was transferred to a new tube and then 20 μL of 3 M NaOAc (pH 5.3) and 50 μL of 95% ethanol were added and precipitated for 2 h at −20 °C. Then, the sample was centrifuged at 12,000× *g* for 20 min at 4 °C to discard the supernatant. The pellet was washed with 500 μL of 95% ethanol by centrifugation at 12,000× *g* for 5 min at 4 °C. The precipitated RNA was dried in a laminar flow cabinet for 5–10 min, resuspended with 50 μL of 1× RNA safeguard reagent (Genemark, Taipei City, Taiwan), and treated with 0.1 U μL^−1^ DNase I Amp Grade (Invitrogen) per μg of total RNA to remove DNA. The quality of total RNA was checked on a 1% agarose gel using the ratio of OD_260_ to OD_280_.

### 4.10. Extraction of Yeast Total Proteins, Soluble Proteins, and Membrane Proteins

Yeast cell pellets (0.25–0.5 g per wet weight) were resuspended in 1 mL ice-cold homogenization buffer containing 50 mM Tris-HCl (pH 7.5), 1 mM EDTA, 1% β-mercaptoethanol, 10% glycerol, and 1 mM phenylmethanesulfonylfluoride (PMSF). Glass beads (1 mm size) were added, and the cell suspension was then vortexed three times for 5 min with 3 min breaks between cycles (4 °C). The glass beads were then sedimented by placing the homogenized suspension on ice. The upper layer suspension was transferred to a fresh pre-cooled Eppendorf tube and centrifuged at 1600× *g* for 5 min to pellet the unbroken cells and cell debris, and the supernatant was considered to be the total protein fraction. The total protein fractions were centrifuged at 18,000× *g* for 1 h into sediment total membrane, and the supernatant was considered to be the soluble protein fraction. The pellet consisting of sedimented crude membranes was resuspended in 1 volume of solubilization buffer containing 50 mM Tris-HCl (pH 7.5), 400 mM NaCl, 1% β-mercaptoethanol, 20% glycerol, and 1 mM PMSF, in which the 400 mM NaCl was permitted to strip peripheral membrane proteins. The crude membranes were then harvested by centrifugation at 18,000× *g* for 1 h. The final pellets containing the crude total membranes were resuspended in solubilization buffer and stored at −20 °C.

### 4.11. Measurement of the Protein Concentration

The protein concentration measurement was carried out according to the method of Bradford [82]. BSA standards (0.1, 0.2, 0.4, 0.6, and 0.8 mg mL^−1^) were added as 20 μL aliquots to 1 mL of Quick Start Bradford protein assay dye (Bio-Rad, Hercules, CA, USA) and incubated at room temperature for 5 min. Absorbances of each standard and sample were read at 595 nm in a spectrophotometer (Jasco V-530, Tokyo, Japan). The absorbance of the standards versus their concentration was plotted. The concentration of the unknown protein was calculated according to the extinction coefficients.

### 4.12. Fluorescence Detection

The fluorescence of EGFP was detected to examine whether the MT-EGFP, MT-Nad1-EGFP, MT-Nad2-EGFP, MT-Nad4-EGFP, and MT-Nad5-EGFP proteins were successfully expressed in yeast mitochondria. MitoTracker Red FM (M22435, Invitrogen) was used as a mitochondrial marker stain. The yeast cells were harvested and resuspended with 10 mM 4-(2-hydroxyethyl)-1-piperazineethanesulfonic acid (HEPES) buffer (pH 7.4) and 5% glucose at a concentration of 10^6^ cells mL^−1^. MitoTracker Red FM was added to the cell suspension to a final concentration of 100 nM and incubated at room temperature for 15–30 min. The fluorescence was observed under a fluorescence microscope (DP71, Olympus, Tokyo, Japan). Bright-field microscopy was used to monitor the integrity of the cells and the same cells were photographed in fluorescence with a GFP filter set (excitation λ_max_: 488 nm, emission λ_max_: 509 nm) and then with the Mito Tracker red filter set (excitation λ_max_: 581 nm, emission λ_max_: 644 nm).

### 4.13. Western Blot Analysis

The total mitochondrial proteins were first separated with sodium dodecyl sulfate polyacrylamide gel electrophoresis (SDS-PAGE) following the method of Laemmli [86]. Then, the protein bands on the SDS-PAGE gel were electro-transferred to a Hybond P polyvinylidene difluoride (PVDF) membrane (GE Healthcare Life Science, formerly Amersham Bioscience, Marlborough, MA, USA) at 400 mA using blot-transfer buffer containing 25 mM Tric-HCl (pH8.3), 192 mM glycine, and 20% (*v*/*v*) methanol in a Hofer TE22 transfer tank (GE Healthcare Life Science, formerly Amersham Pharmacia Biotech, Marlborough, MA, USA) at 4 °C for 90 min. After transfer, the membranes were washed with phosphate-buffered solution (PBS) containing 80 mM Na_2_HPO_4_ (pH 7.5), 20 mM NaH_2_PO_4_·H_2_O, and 100 mM NaCl for 5 min. The PVDF membranes were subsequently blocked in PBS buffer containing 5% non-fat milk at 4 °C overnight. The following day, the PVDF membranes were probed with anti-GFP (1:2000 dilution, AB3080, Millipore, now part of Merck, Darmstadt, Germany) primary antibodies in PBS containing 1% (*v*/*v*) non-fat milk at 4 °C overnight. These steps were followed by three cycles of 15 min washes with PBS plus 0.05% (*v*/*v*) Tween 20. Then, the PVDF membranes were probed with anti-rabbit IgG horseradish peroxidase (HRP) conjugate (1:2000 dilution) secondary antibody in PBS containing 1% (*v*/*v*) non-fat milk for 1 h at room temperature. After a further three cycles of 15 min washes with PBS plus 0.05% (*v*/*v*) Tween 20, the PVDF membranes were detected using DAB solution and a 2.5 mg diaminobenzidine (DAB, Merck) tablet dissolved in 100 mL of ddH_2_O with 28 μL of 30% (*v*/*v*) H_2_O_2_ was added. The PVDF membranes were stained for 5 to 10 min. The reaction was stopped as soon as the specifically stained bands were clearly visible, and the PVDF membrane was washed three times with ddH_2_O. Finally, the PVDF membrane was scanned with a ScanMarker 9800 XL scanner (Microtex, Hsinchu City, Taiwan).

### 4.14. Isolation of Yeast Mitochondria

The yeast mitochondria were isolated according to the method of Meisinger et al. [87] with some modifications. The cells were harvested at room temperature by centrifugation at 3000× *g* for 5 min and washed with distilled water. The cells were resuspended in pre-warmed dithiothreitol (DTT) buffer (2 mL per gram of wet weight cells) containing 0.1 M Tris-H_2_SO_4_ (pH 9.4) and 10 mM DTT, and shaken at 80 rpm at 30 °C for 20 min. The cells were harvested by centrifugation at 3000× *g* for 5 min and resuspended in Lyticase buffer containing 0.5 mg Lyticase (Sigma-Aldrich; 7 mL per gram of wet weight cells), 1.2 M sorbitol, and 20 mM KH_2_PO_4_ (pH 7.4). Then, the cells were harvested by centrifugation at 3000× *g* for 5 min, resuspended in Lyticase buffer (7 mL per gram of wet weight cells), and shaken at 80 rpm at 30 °C for 3–4 h. The spheroplasts of lytic cells were harvested by centrifuging at 3000× *g* for 5 min, washed with Lyticase buffer (7 mL per gram of wet weight cells), and resuspended in ice-cold homogenization buffer (6.5 mL per gram of wet weight cells) containing 600 mM sorbitol, 1 mM EGTA, 20 mM 2-(*N*-morpholino)ethanesulfonic acid (MES)-KOH (pH 6.0), 0.1% BSA, and 0.5 mM PMSF. The cell suspensions from this step were maintained at low temperature to avoid proteolysis. Then, the cell suspensions were transferred into a 100 mL homogenizer to homogenize the spheroplasts with 15 strokes. The homogenate was diluted twofold with homogenization buffer and centrifuged at 1500× *g* for 5 min at 4 °C into pellet cell debris and nuclei. The supernatant was centrifuged at 4000× *g* for 5 min at 4 °C to discard the pellet. The collected supernatant was centrifuged at 12,000× *g* for 15 min. Finally, the crude mitochondrial pellet was resuspended in a solubilization buffer containing 0.6 M sorbitol, 1 mM EGTA, and 20 mM HEPES-KOH (pH 7.4). The mitochondrial protein concentration was determined using the method of Bradford [74]. The sample, which had a final concentration of 5–10 mg proteins mL^−1^, was stored at −80 °C.

### 4.15. Influence of Salt or pH on the Growth of S. cerevisiae

*S. cerevisiae* transformants were grown overnight to saturation in liquid SC-U medium plus 2% glucose, washed with ddH_2_O, and resuspended in SC-U medium plus 2% galactose to induce the expression of MT-Nad1-EGFP, MT-Nad2-EGFP, MT-Nad4-EGFP, or MT-Nad5-EGFP fusion proteins. Tenfold serial dilutions of cell suspension (10^−1^ to 10^−5^) were prepared and spotted in 3 μL aliquots onto SC-U medium plus 2% galactose agar plates. In the salt-dependent experiment, 100 mM LiCl, 600 mM NaCl, or 800 mM KCl were added [39]. In the pH-dependent experiment, the pH values of the media were adjusted to pH 4.5, pH 5.5, or pH 6.5. Cell growth was monitored for 3 days at 30 °C.

## Figures and Tables

**Figure 1 ijms-23-04054-f001:**
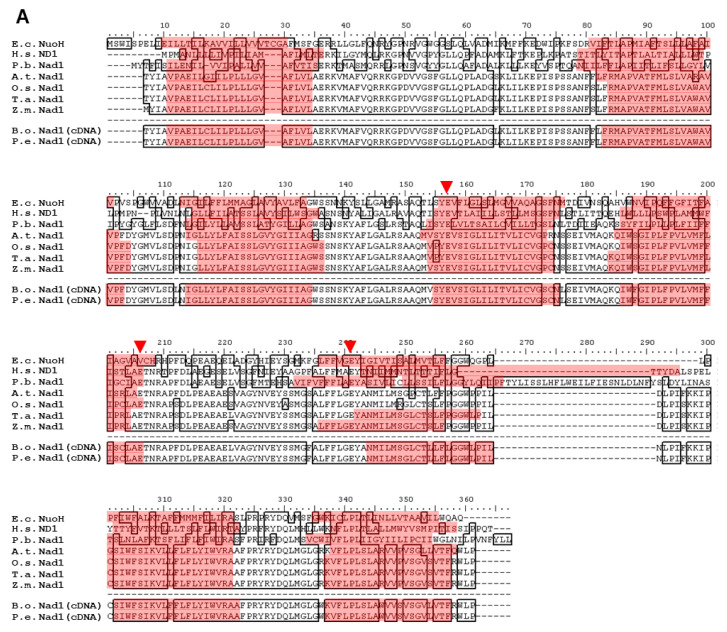
Comparison of bamboo mitochondrial Nad1, Nad2, Nad4, and Nad5 amino acid sequences with the homologous proteins in other species. The polypeptide sequences of Nad1 (**A**), Nad2 (**B**), Nad4 (**C**), and Nad5 (**D**) translated from the cDNA of *B. oldhamii* (B.o.; ABZ79354, KF219802, ABZ79355, and ABZ79356) and Nad1, Nad4, and Nad5 from that of *P. edulis* (P.e.; ABV68729, ABZ79357, and ABZ79358) compared with those of their homologs from *E. coli* (E.c.; BAA16110, BAA16113, BAA16105, and BAA16106), *H. sapiens* (H.s.; AAB58943, AAB58944, AAB58952, and AAB58953), *P. brasiliensis* (P.b.; AAY30326, AAY30327, AAY30329, and AAY30339), *A. thaliana* (A.t.; CAA69811, CAA69771, CAA69742, and CAA69752), *O. sativa* (O.s.; AAZ99270, AAZ99285, AAZ99271, and AAZ99286), *T. aestivum* (T.a.; BAE47688, BAE47677, BAE47673, and BAE47687), and *Z. mays* (Z.m.; AAR91201, AAR91195, AAR91196, and AAR91197), as well as the sequence translated from *B. oldhamii* mtDNA (B.o.; ABY55185), using BioEdit software. The accession numbers of Nad1, Nad2, Nad4, and Nad5 were obtained from the NCBI database. The putative transmembrane domains of each subunit were predicted by ConPred II and are indicated in the red regions. The regions marked in blue are probably α-helices.

**Figure 2 ijms-23-04054-f002:**
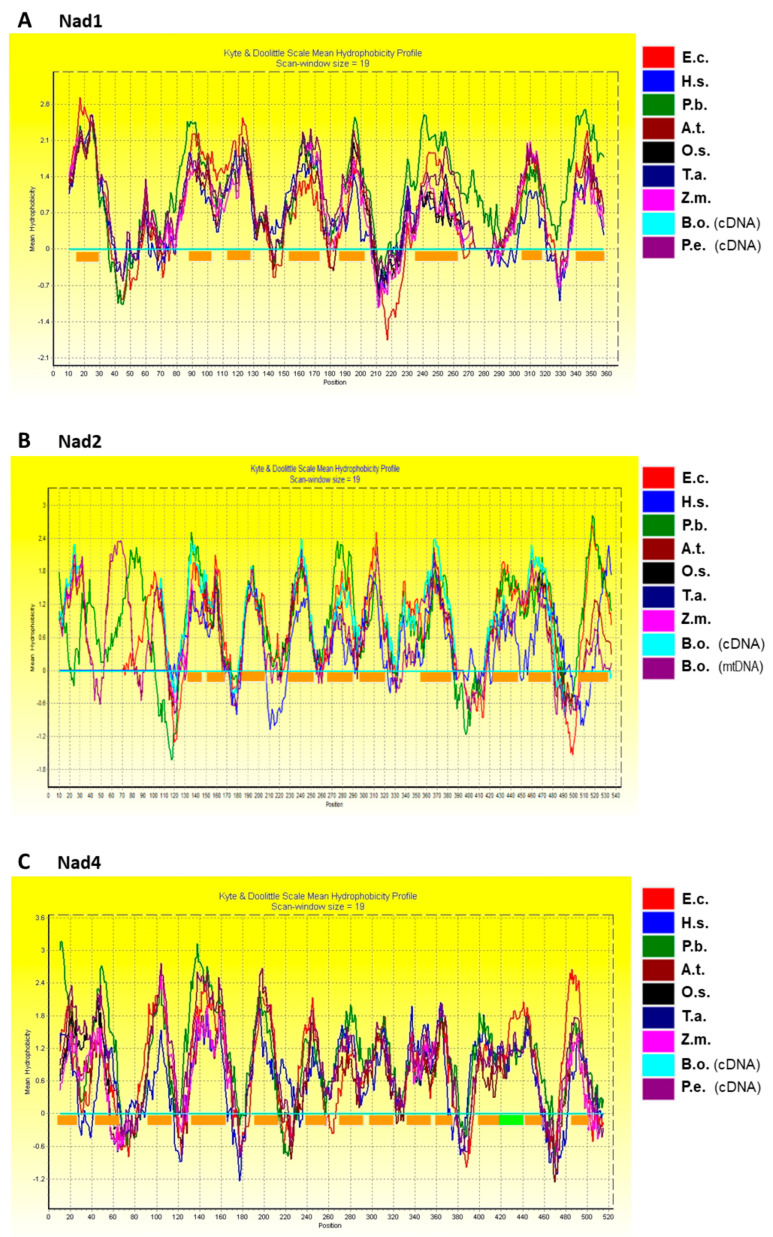
Hydropathy plots of bamboo Nad1, Nad2, Nad4, and Nad5 subunits compared to their homologous proteins in other species. *B. oldhamii* and *P. edulis* mitochondrial Nad1 (**A**), Nad2 (**B**), Nad4 (**C**), and Nad5 (**D**) were compared with *E. coli* (E.c.; BAA16110, BAA16113, BAA16105, and BAA16106), *H. sapiens* (H.s.; AAB58943, AAB58944, AAB58952, and AAB58953), *P. brasiliensis* (P.b.; AAY30326, AAY30327, AAY30329, and AAY30339), *A. thaliana* (A.t.; CAA69811, CAA69771, CAA69742, and CAA69752), *O. sativa* (O.s.; AAZ99270, AAZ99285, AAZ99271, and AAZ99286), *T. aestivum* (T.a.; BAE47688, BAE47677, BAE47673, and BAE47687), and *Z. mays* (Z.m.; AAR91201, AAR91195, AAR91196, and AAR91197). The accession numbers of Nad1, Nad2, Nad4, and Nad5 were obtained from the NCBI database as described in Figure 1. The horizontal thick lines in orange indicate the α-helices and transmembrane domains.

**Figure 3 ijms-23-04054-f003:**
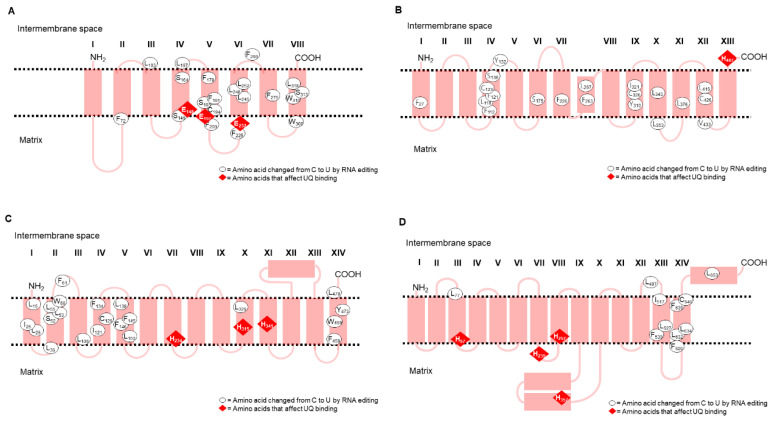
Transmembrane topology models of bamboo mitochondrial Nad1, Nad2, Nad4, and Nad5. The transmembrane domains of the (**A**) Nad1, (**B**) Nad2, (**C**) Nad4, and (**D**) Nad5 polypeptide sequences from *B. oldhamii* and *P. edulis* were predicted using the website ConPred II. Amino acids that were changed by C-to-U RNA editing are shown in ovals. The amino acids that may affect UQ binding are shown in red diamonds.

**Figure 4 ijms-23-04054-f004:**
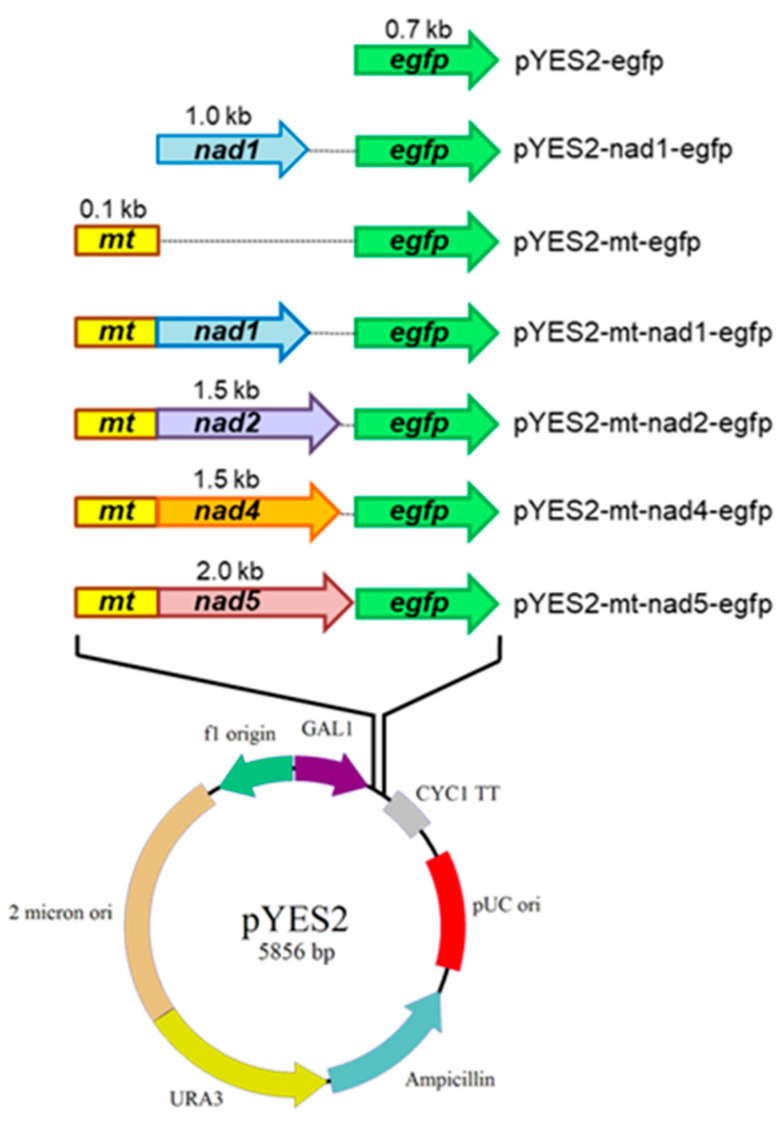
Schematic diagram of the constructed plasmids transformed into *S. cerevisiae*. The mitochondrion-encoded *nad1*, *nad2*, *nad4*, and *nad5* cDNAs were inserted into the pYES2-mt-egfp plasmid. *mt*: mitochondrial targeting gene; *egfp*: enhanced green fluorescent protein gene; GAL1: yeast GAL1 promoter; CYC1 TT: transcription terminator; pUC ori: E. coli replication origin bases; ampicillin: ampicillin resistance gene; URA3: orotidine 5-phosphate decarboxylase gene; 2 micron ori: maintenance and high-copy replication origin in yeast; f1 origin: F1 phage origin.

**Figure 5 ijms-23-04054-f005:**
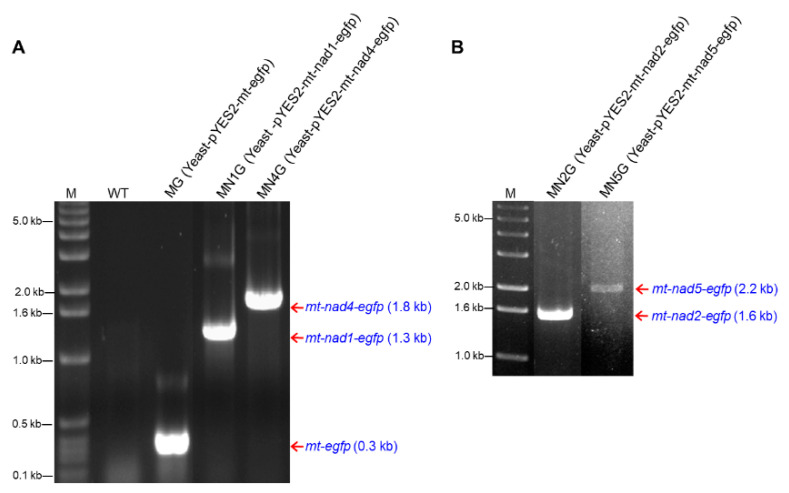
RNA transcription of fusion genes in yeast transformants. Reverse transcription PCR from total RNAs of yeast transformants that were amplified by using a pair of primers, mt-egfpF and mt-egfpR. (**A**) The 0.3 kb *mt-egfp* fragment, the 1.3 kb *mt-nad1-egf* fragment, and the 1.6 kb *mt-nad2-egfp* fragment. (**B**) The 1.8 kb *mt-nad4-egfp* fragment and the 2.2 kb *mt-nad5-egfp* fragment were expected as products. WT: yeast wild-type; MG: yeast-pYES2-mt-egfp transformant; MN1G: yeast-pYES2-mt-nad1-egfp transformant; MN2G: yeast-pYES2-mt-nad2-egfp transformant; MN4G: yeast-pYES2-mt-nad4-egfp transformant; MN5G: yeast-pYES2-mt-nad5-egfp transformant; M: 0.1 μg 1 kb DNA ladder marker.

**Figure 6 ijms-23-04054-f006:**
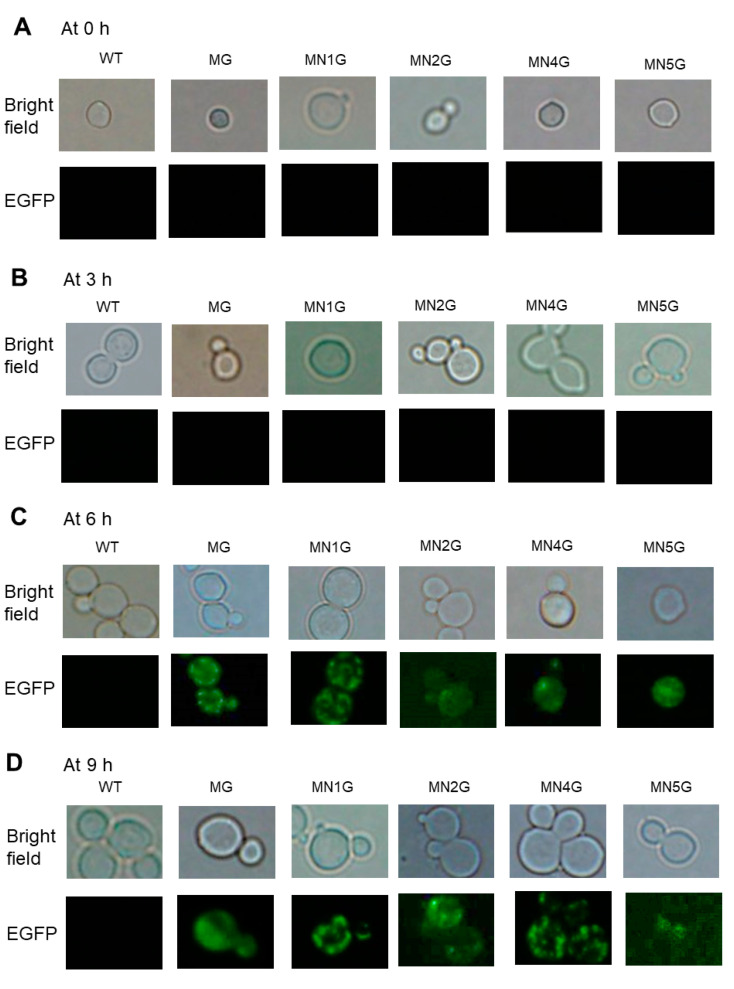
Time course of MT-Nad1-EGFP, MT-Nad2-EGFP, MT-Nad4-EGFP, and MT-Nad5-EGFP fusion proteins in yeast observed by fluorescence microscopy. The cells of yeast WT and transformants were induced with SC-U medium plus 2% galactose for 0–9 h at 30 °C to generate an EGFP signal. The images were taken at (**A**) 0 h, (**B**) 3 h, (**C**) 6 h, and (**D**) 9 h. Fluorescence microscopy was used to detect EGFP, which showed green fluorescence in the mitochondria. Bright-field microscopy was used to monitor the integrity of the cells.

**Figure 7 ijms-23-04054-f007:**
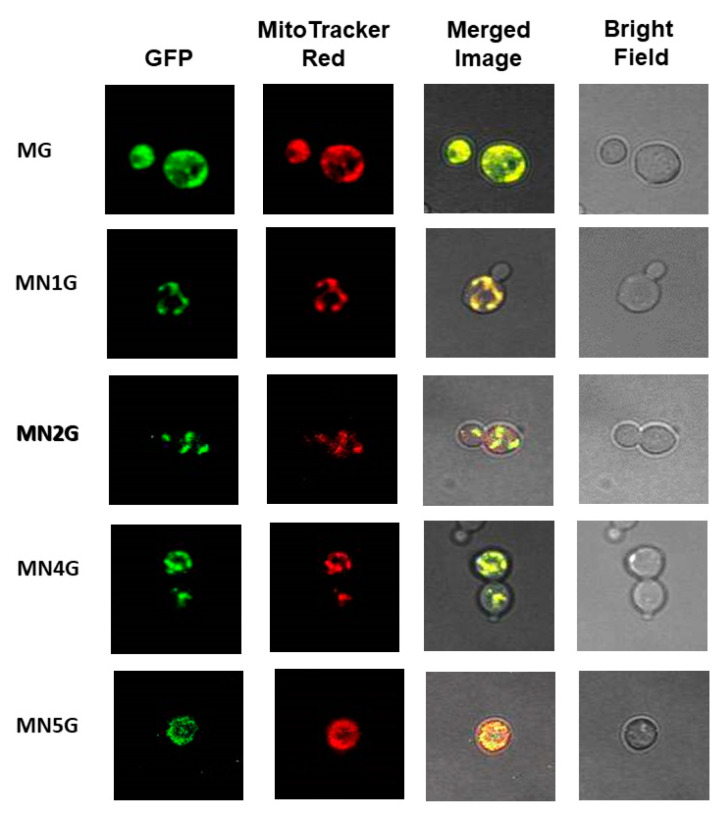
Subcellular localization of MT-EGFP, MT-Nad1-EGFP, MT-Nad2-EGFP, MT-Nad4-EGFP, and MT-Nad5-EGFP fusion proteins after expression in yeast observed with fluorescence microscopy. Yeast cells of MG, MN1G, MN2G, MN4G, and MN5G were incubated in SC-U medium plus 2% galactose for 15 h at 30 °C to generate an EGFP signal. Green fluorescence indicates the mitochondria containing EGFP, red fluorescence indicates the mitochondria labeled with MitoTracker Red, and the yellow regions in the merged images indicate the co-occurrence in the mitochondria. The integrity of the cells was monitored by bright-field microscopy.

**Figure 8 ijms-23-04054-f008:**
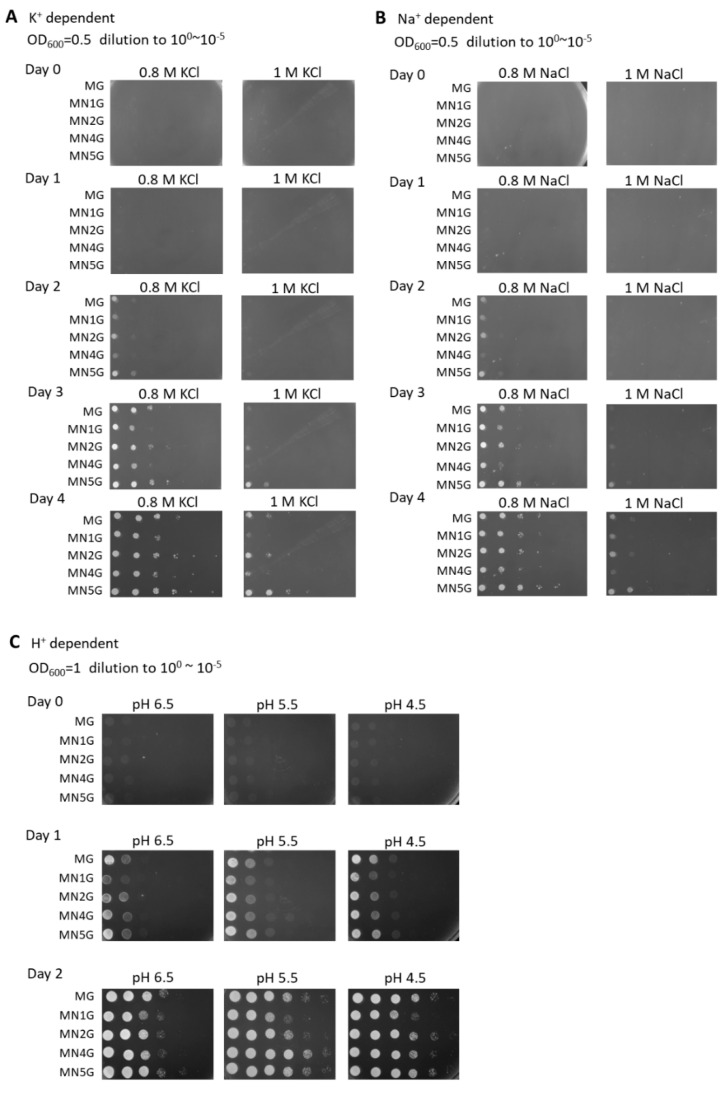
K^+^, Na^+^, or H^+^-dependent growth phenotypes of yeast transformants. (**A**) Yeast transformants were cultured to saturation for 15 h in 3 mL SC-U medium plus 2% glucose and diluted from OD_600_ = 0.5, and were then pipetted as a 1:10 dilution series onto SC-U medium plus 2% galactose agar plates with the addition of 0.8 M NaCl or 1 M NaCl. The white colonies indicate 10^0^~10^−5^ series diluted cells from left to right and were observed from day 0 to day 4. (**B**) Yeast transformants were cultured and treated as above except with the addition of 0.8 M KCl or 1 M KCl. The white colonies indicate 10^0^~10^−5^ series diluted cells from left to right and were observed from day 0 to day 4. (**C**) Yeast transformants were cultured to saturation for 15 h in 3 mL SC-U medium plus 2% glucose and diluted from OD_600_ = 1, and then were pipetted as a 1:10 dilution series onto SC-U medium plus 2% galactose agar plates at pH 6.5, pH 5.5, or pH 4.5. The white colonies indicate 10^−1^~10^−6^ diluted cells from left to right and were observed from day 0 to day 2.

## Data Availability

Not applicable.

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
