# Peer review of "Cloning and Organelle Expression of Bamboo Mitochondrial Complex I Subunits Nad1, Nad2, Nad4, and Nad5 in the Yeast Saccharomyces cerevisiae"

_ijms, 2022, doi:10.3390/ijms23074054_

Round 1
Reviewer 1 Report
Revision of the Manuscript Cloning and organelle expression of bamboo mitochondrial 2 complex I subunits Nad1, Nad2, Nad4, and Nad5 in the yeast 3 Saccharomyces cerevisiae.
The manuscript by Tsai et al. describes the heterologous expression of bamboo mitochondrial complex I subunit genes nad1, nad2, nad4, and nad5 in S. cerevisiae. The gene products reside at the inner membrane and possibly function as proton and ion translocators. Gene transcription was monitored by measuring RNA transcript levels. Expression and correct localization of the proteins was proven by fluorescence microscopy by fusing them to a mitochondrial targeting signal sequence and adding an EGFP tag. Growth phenotype assays on media containing different concentrations of K+, Na+, or H+ suggest that Nad1, Nad2, Nad4, and Nad5 may function in ion translocations.
Comments
Major Revisons
The article is generally well-constructed and the results are presented in a convincing way. However, there are several experimentally relevant controls missing. Also, most fluorescence microscopy images are of very bad quality, which has to be improved in order to be accepted. The language could to be improved.
Major remarks:
- The introduction misses an explanation, why cerevisiae lacks mitochondrial complex I. This is essential to understand, in how far Nad proteins can even have a functional role in yeast, since it naturally does not express any of these proteins.
- Figure 2: Figure Description is not clearly understandable. Please indicate what ‘The thick lines indicate the α-helices and trans-membrane domains.’ means. Color code? Orange and green parts?
- Figure 6: Many cells are out of focus. Also, cells of similar size and growth stage should be chosen for comparison.
- Figure 7: Many cells are out of focus, which makes cellular co-localization of Nad poteins with MitoTracker almost impossible. Specifically, MG and MN1G are extremely out of focus. Also, it looks as if cells picked for MG and MN5G are dead cells, as judged from their shrunk nature and the fact that the whole cell interior seems to fluoresce.
- Figure 8: Spot growth assays lack SC-U control plates w/o any addition of salts etc. to compare general growth of different strains. Is there a general growth phenotype? E.g. MN1G seems to grow slower than the wild-type on all plates shown.
Minor remarks:
- Last sentence of abstract has a couple of extra words who need to be deleted. (‘which that were was’)
Reviewer 2 Report
The manuscript reports a profound study on cloning and sequence-level characterization of cDNA sequences of the Mitochondrial Complex I in two bamboo species, as well as on their heterologous expression and mitochondrial targeting in yeast. The results are convincingly presented and the manuscript reads well. I have no major concerns regarding the content of the manuscript. However, in my opinion the general parts (Abstract, Introduction, Discussion) could be better organized in respect to the wider audience. For instance, to insert a short paragraph at the end of the Introduction about the main goals and objectives of the study would be beneficial. Among other benefits, this would also help to clarify that two local bamboo species/varieties were chosen for the studies - as in the present version of manuscript these two species are first mentioned in the title of the first subsection of the Results section (Line 120). On the other hand, the Discussion is too long and a great deal of it is just repeating facts shown in the Results part. The authors should try to make the Discussion more compact and better structured and emphasize the most significant findings and results of the study. There is a very short Conclusions chapter at the very end (after the Methods chapter ...) of the manuscript. This can be
merged into the Discussion (or simply deleted, as this one-sentence "chapter" does not tell too much ...). The language of the manuscript is generally
appropriate, but a final proofreading/styling would be beneficial. Apart from some (mostly) minor style- and wording weaknesses, there is at least one type of wording that definitively needs correction: The authors repeatedly use the term "splicing" for nucleotide changes resulted from RNA editing (Line 169: "The remaining 4.3% underwent C-to-U splicing during RNA editing..." Line 426: "...while the remaining 27 % comprised C-T splicing ..."). In molecular biology the term "splicing" is generally (and exclusively) interpreted as a process of joining exons and "splicing off" introns from the pre-mRNA, while RNA editing (which is common in mitochondrial transcripts) results in nucleotide substitutions.
